# Video-Helpful Multimodal Machine Translation

**Yihang Li[1], Shuichiro Shimizu[1], Chenhui Chu[1], Sadao Kurohashi[1], Wei Li[2]**

[1]Kyoto University, [2]Google Research

{liyh, sshimizu, chu, kuro}@nlp.ist.i.kyoto-u.ac.jp, mweili@google.com

## Abstract

Existing multimodal machine translation (MMT) datasets consist of images and video captions or instructional video subtitles, which rarely contain linguistic ambiguity, making visual information ineffective in generating appropriate translations. Recent work has constructed an ambiguous subtitles dataset to alleviate this problem but is still limited to the problem that videos do not necessarily contribute to disambiguation. We introduce EVA (Extensive training set and Video-helpful evaluation set for Ambiguous subtitles translation), an MMT dataset containing 852k Japanese-English (Ja-En) parallel subtitle pairs, 520k Chinese-English (Zh-En) parallel subtitle pairs, and corresponding video clips collected from movies and TV episodes. In addition to the extensive training set, EVA contains a video-helpful evaluation set in which subtitles are ambiguous, and videos are guaranteed helpful for disambiguation. Furthermore, we propose SAFA, an MMT model based on the Selective Attention model with two novel methods: Frame attention loss and Ambiguity augmentation, aiming to use videos in EVA for disambiguation fully. Experiments on EVA show that visual information and the proposed methods can boost translation performance, and our model performs significantly better than existing MMT models. The EVA dataset and the SAFA model are available at: https://github.com/ku-nlp/video-helpful-MMT.git.

## 1 Introduction

Neural machine translation (NMT) (Bahdanau et al., 2015; Wu et al., 2016) models relying on text data have achieved state-of-the-art performance. However, in many cases, the text is insufficient to provide the information needed for appropriate translation, especially when the source text is ambiguous. In this work, "Ambiguous" refers not only to ambiguity in a narrow sense caused by

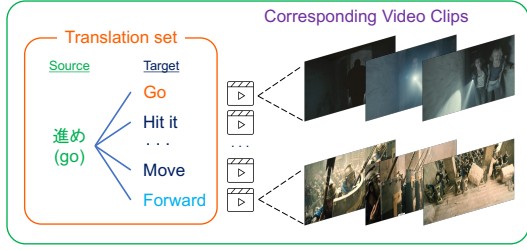

Figure 1: A translation set and the corresponding video clips. We combine parallel subtitles with source subtitles in Japanese and target subtitles in English into a translation set.

factors such as polysemy but also to different possible translations caused by factors such as emotion, politeness, and omission that needs multimodal information for disambiguation. For a source text, if there are multiple possible translations and some of them are more appropriate than others under certain visual scenes, we call it ambiguous. Multimodal machine translation (MMT) (Specia et al., 2016; Sulubacak et al., 2020a) uses visual data as auxiliary information to tackle the ambiguity problem. The contextual information in the visual data helps to resolve the ambiguity in the source text data.

Previous MMT studies have mainly focused on the image-guided machine translation (IMT) task (Elliott et al., 2016; Zhao et al., 2020; Li et al., 2022a), where, given an image and a source sentence, the goal is to enhance the quality of translation by leveraging their semantic correspondence to the image. Resolving ambiguities through visual cues is one of the main motivations behind this task. Compared with images, videos contain ordered sequences of frames and can provide richer visual features such as motion features. Recently, some studies have started to focus on the video-guided machine translation (VMT) task (Wang et al., 2019; Gu et al., 2021).

VMT faces the problem of data scarcity. The How2 (Sanabria et al., 2018) and VATEX (Wang et al., 2019) datasets are recent efforts to allevi-

ate the problem. In addition, previous datasets are limited to subtitles of instructional videos or video captions that describe the video clips. It has been shown that caption MMT essentially does not require visual information due to the lack of language ambiguity in captions (Caglayan et al., 2019). At the same time, the subtitles of the instructional videos are similar to captions and also lack ambiguity. The VISA (Li et al., 2022b) dataset has made efforts to solve this problem. VISA contains parallel subtitles and corresponding video clips collected from movies or TV episodes in which the source subtitles are ambiguous. However, it has a limitation that the videos do not necessarily contribute to disambiguation.

To address the problems of previous VMT datasets, we construct a new large-scale VMT dataset EVA (Extensive training set and Video-helpful evaluation set for Ambiguous subtitles translation) for VMT research. EVA has an extensive training set, and a video-helpful evaluation set in which source subtitles are ambiguous and videos are guaranteed to be helpful for disambiguation. In total, EVA contains 852k Ja-En parallel subtitle pairs, 520k Zh-En parallel subtitle pairs, and corresponding video clips collected from movies and TV episodes, where each pair of parallel subtitles has a corresponding video clip. Subtitles from movies and TV episodes are essentially dialogues and short (7.34 English words in our case), which makes subtitles have many possible interpretations and thus makes videos helpful.

To train a VMT model that can disambiguate translations, it is necessary for the training set to contain possible translation patterns. To achieve this, a simple yet effective way is to make the training set as large as possible. The training set of EVA contains 848k Ja-En parallel subtitle pairs, 517k Zh-En parallel subtitle pairs, and corresponding video clips, which are collected from 763 movies and 1,361 TV episodes with a total length of 3,791 hours. The training set is much larger than existing VMT datasets and may cover more subtitle translation patterns.

For testing VMT models, it is inappropriate to use general data because many translations do not require visual information. Therefore, we select video-helpful data to construct an evaluation set that contains 4,276 Ja-En parallel subtitle pairs, 2,940 Zh-En parallel subtitle pairs, and corresponding video clips. Considering the definition of am-

biguity and the need for the training set containing possible translation patterns, we collect video-helpful data using *translation sets*, which are sets of parallel subtitles that have the same source subtitles but different target subtitles. An example of a translation set is shown in Figure 1. Each pair of parallel subtitles belongs to a video clip. As a translation task, the video clip can help us translate the source subtitle. The first video clip shows two women escaping, suggesting a "go" translation, while the last video clip portrays a scene involving an army, suggesting a "forward" translation. We construct the dataset by collecting parallel subtitles and corresponding video clips, constructing an evaluation set with translation sets and crowd-sourcing, and then using the remaining part as the training set.

Furthermore, we propose an MMT model SAFA (Selective Attention model with Frame attention loss and Ambiguity augmentation) for the VMT task. Based on the selective attention model proposed (Li et al., 2022a) for the IMT task, we 1) use CLIP4Clip (Luo et al., 2022) model to extract video features, 2) propose frame attention loss to make the model focus more on the central frames where the subtitles occur, and 3) propose ambiguity augmentation to make the model put more weights on the possibly-ambiguous data. Experiments on EVA show that SAFA achieves 15.41 BLEU score and 35.86 METEOR score for Ja-En translation, and 27.62 BLEU score and 48.74 METEOR score for Zh-En translation, which are significantly better than existing MMT models. Furthermore, our proposed methods significantly improve MT performance when incorporating videos, with relative improvements of 9.99% and 4.94% in BLEU scores for Ja-En and Zh-En translations, respectively.

In summary, our contributions are three-fold:

- We construct EVA, a large-scale parallel subtitles and video clips dataset, to promote VMT research.

- We propose the SAFA model with frame attention loss and ambiguity augmentation.

- We conduct substantial experiments on the EVA dataset with SAFA to set a benchmark of the dataset.

## 2   Related Work

**Multimodal Machine Translation.**   MMT involves drawing information from multiple modal-

| Dataset | Domain | Language | Duration | #sent | #video |
|---------|--------|----------|----------|-------|--------|
| How2 | instruction | En-Pt | 90s | 189,276 | 13,662 |
| VATEX | caption | En-Zh | 10s | 349,910 | 34,991 |
| VISA | subtitle | Ja-En | 10s | 39,880 | 39,880 |
| EVA (Ours) | subtitle | Ja-En, Zh-En | 10s | 1,372,113 | 1,372,113 |

Table 1: Statistics of VMT datasets. #sent stands for the number of parallel sentences. The duration of videos in the How2 dataset is average duration.

ities, assuming that they should contain useful alternative views of the input data (Sulubacak et al., 2020b). Previous studies mainly focus on IMT using images as a visual modality to help machine translation (Specia et al., 2016; Elliott et al., 2017; Barrault et al., 2018; Su et al., 2019; Yang et al., 2020; Li et al., 2022a). The usefulness of the visual modality has recently been disputed under specific datasets or task conditions (Elliott, 2018; Caglayan et al., 2019; Wu et al., 2021). However, using images in captions translation is theoretically helpful for handling grammatical characteristics and resolving ambiguities when translating between dissimilar languages (Sulubacak et al., 2020b). VMT is a MMT task similar to IMT but focuses on video clips rather than images associated with the textual input. Existing VMT models mainly focus on video caption translation and use video motion features extracted with the I3D (Carreira and Zisserman, 2017) model to help translation (Wang et al., 2019; Gu et al., 2021). The hierarchical attention network (Gu et al., 2021) has been proposed further to combine motion features, and object features (Ren et al., 2015).

**VMT Datasets.** The scarcity of datasets is one of the largest obstacles to the advancement of VMT. Recent efforts to compile freely accessible data for VMT, such as the How2 (Sanabria et al., 2018), VATEX (Wang et al., 2019) and VISA (Li et al., 2022b) datasets, have begun to alleviate this bottleneck. Table 1 shows the statistics of existing VMT datasets and EVA. EVA is the largest VMT dataset in video hours and a number of video clips and parallel sentences. Very recently, the BigVideo (Kang et al., 2023) dataset consisting of 4.5 million Zh-En sentence pairs and 9,981 hours of YouTube videos is proposed to facilitate the study of MMT. We consider this work contemporaneous to our study. The evaluation set of BigVideo is annotated by professional speakers in both Chinese and English to enhance the quality, while we design a language-independent pipeline with translation sets to ensure the scalability of the evaluation set and reusability

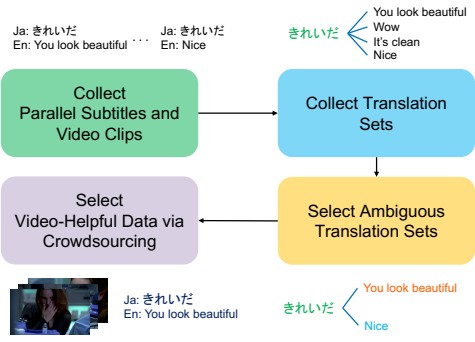

Figure 2: The pipeline of dataset construction.

of the pipeline.

## 3 Dataset

In this section, we outline the construction of EVA, which is a VMT dataset combined with parallel subtitles and corresponding video clips. The dataset contains a large-scale training set and a small video-helpful evaluation set. For the former, we make it large enough to cover different translation patterns. For the latter, we ensure that the source subtitles are ambiguous and the video clips can help disambiguate the source subtitles.

### 3.1 Pipeline

To construct the dataset, we first collect a large number of parallel subtitles and one-by-one corresponding video clips, where each pair of parallel subtitles matches a video clip. For these data, we select some video-helpful data to construct the evaluation set and use the remaining data as the training set. To select video-helpful data, we collect translation sets from the parallel subtitles, select ambiguous translation sets, and do crowdsourcing to further select video-helpful data. We construct the evaluation set with the goal of reaching a specific number instead of covering all video-helpful data. In this way, we can both get a video-helpful evaluation set and keep the most ambiguous data in the training set. As the dataset construction pipeline is language-independent, it can be extended to other language pairs. The pipeline is shown in Figure 2.

### 3.1.1 Collect parallel subtitles and video clips

We collect parallel subtitles and corresponding video clips following the method in (Li et al., 2022b). On the one hand, the collected data can be used to construct an extensive training set. On the other hand, the collected data can be used to further extract the video-helpful evaluation set.

Regarding the parallel subtitles, we collect Japanese–English and Zh-En parallel subtitles from the OpenSubtitles(Lison and Tiedemann, 2016) dataset. OpenSubtitles is a subtitles dataset compiled from an extensive database of film and TV subtitles which includes a total of 1,689 bitexts spanning 2.6 billion sentences across 60 languages. From OpenSubtitles, we can also collect subtitle timestamps and the Internet Movie Database (IMDb) ids of the video sources.

To collect corresponding video clips, we fix subtitle timestamps and crop video clips according to these timestamps. More details can be found in Appendix B.1. Based on accurate timestamps, we crop 10-second 25-fps video clips for parallel subtitles following (Wang et al., 2019; Li et al., 2022b). From the midpoint of each subtitle's period, each video clip takes 5 seconds before and after, respectively. The audios of most videos are in English, which is the same as the translation target language and may interfere with translation. Therefore, we only keep the video content of video clips and remove the audio content.

As a result, we collected 852,440 Ja-En parallel subtitles, 519,673 Zh-En parallel subtitles, and corresponding video clips. This way, we can construct a VMT training set much more extensive than existing VMT datasets.

### 3.1.2 Collect translation sets

After collecting parallel subtitles and corresponding video clips, we can collect translation sets from the parallel subtitles. Translation sets are sets of parallel subtitles that have the completely same source subtitles but different target subtitles (i.e., translations). Therefore, a large number of parallel subtitles is a prerequisite for collecting translation sets. Note that the parallel subtitles with similar but not completely the same source subtitles are not collected into the translation sets and therefore retained in the training set. In this step, we collected 26,533 Ja-En translation sets and 17,642 Zh-En translation sets.

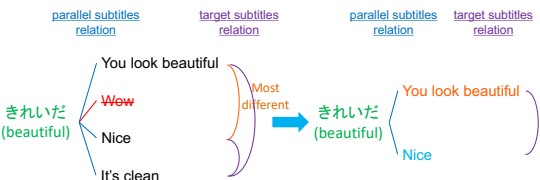

Figure 3: The selection of an ambiguous translation set.

### 3.1.3 Select ambiguous translation sets

Then we select ambiguous translation sets with sentence similarity. An ambiguous translation set contains a source subtitle and two target subtitles with different meanings. As shown in Figure 3, we select ambiguous translation sets considering two points. On the one hand, we ensure the target subtitles have different meanings. On the other hand, we ensure the target subtitles are parallel with the source subtitle. Subtitles in the OpenSubtitles dataset are provided by volunteers and therefore contain some non-parallel subtitles. If we only focus on the first point, we might often select the non-parallel target subtitles because they tend to have a totally different meaning from other target parallel subtitles.

We propose a method to balance the two points discussed above. The main idea is as the following. We first keep the parallel subtitles similarity as high as possible. Then we select the most different target subtitles pair. If they are different enough, we combine them and the source subtitle into an ambiguous translation set and discard the remaining data of the translation set. Otherwise, we relax the restriction on parallel subtitles' similarity and repeat the process above. More details can be found in Appendix B.3. As a result, we collected 10,594 Ja-En ambiguous translation sets and 7,102 Zh-En ambiguous translation sets.

### 3.1.4 Select video-helpful data via crowdsourcing

At last, we do crowdsourcing to further select video-helpful data. In order to determine whether the video can help disambiguate the source subtitles, the best way should be to distribute tasks to workers who can understand both the source and target languages. However, in practice, it is hard to find such workers. So we designed a scheme that only requires workers to be able to understand the target language.

In each task, we show two target subtitles from the same ambiguous translation set and one video

| Split | train | validation | test |
|---|---|---|---|
| #sample (Ja-En) | 848,164 | 2,138 | 2,138 |
| #sample (Zh-En) | 516,733 | 1,470 | 1,470 |
| video-helpful | - | ✓ | ✓ |

Table 2: EVA splits. "sample" denotes a pair of parallel subtitles and a corresponding video clip.

clip that belongs to one of the two subtitles. Then we ask workers if there is any subtitle strongly related to the video content. And we give workers four choices: (1) none of them; (2) only the first subtitle; (3) only the second subtitle; (4) both of them. In this way, if the video content is only strongly related to the corresponding subtitle, we may estimate that according to the video content, the source subtitle can only be translated to the corresponding subtitle instead of the other one.

We did crowdsourcing on Amazon Mechanical Turk. We distribute each task to three workers. Suppose at least two workers agree that the video content is only strongly related to one of the two subtitles, and the subtitle is the corresponding subtitle; in that case, we regard the video clip and corresponding parallel subtitles as video-helpful data. Other techniques to improve crowdsourcing quality can be found in Appendix B.4. As a result, we constructed a video-helpful evaluation set containing $4,276$ Ja-En parallel subtitle pairs, $2,940$ Zh-En parallel subtitle pairs, and corresponding video clips. And we equally divide this set into a validation set and a test set.

We calculated the inter-annotator reliability to evaluate the crowdsourcing results. Krippendorff's alpha (Krippendorff, 2011) is $0.681$ and $0.728$ for Ja-En and Zh-En. Both of them achieve the substantial agreement (Hughes, 2021).

### 3.2 Dataset analyses

Table 2 shows the splits of EVA. The evaluation set contains many kinds of ambiguities. We checked 50 samples. The most frequent causes are omission, emotion, and polysemy, with approximate proportions of 30%, 30%, and 20%. For the remaining, it is difficult to define the causes of ambiguity. Sometimes, instances of ambiguity arise from a combination of various factors, thereby challenging precise classification. For example, "放せ!"" can be translated as both "Let me go!" and "Drop it!." This ambiguity could arise due to the polysemy "放せ!" or it could stem from the omission of the object. And sometimes, it is difficult to tell whether a subtitle is ambiguous by just checking the source subtitles, and the ambiguity can be easier understood by comparing the different target subtitles. For example, "聞こえ (hear) ますか (can) ?" is usually translated into " Can you hear me?". However, if a person is on the phone and this is his first sentence, "Is there anyone?" is more natural.

## 4 Model

We present a new model, SAFA, for VMT based on the EVA dataset. Previous VMT models mainly focus on caption translation based on the existing video caption datasets. In contrast, our model mainly focuses on subtitle translation. We design SAFA based on a selective attention model, propose a frame attention loss to make the model focus on the central frames where the subtitles occur, and use ambiguity augmentation to make the model put more weight on the possibly-ambiguous data. The model overview is shown in Figure 4.

### 4.1 Selective attention model

There are few existing VMT models, and most of them are based on LSTM or GRU models (Wang et al., 2019; Gu et al., 2021). The selective attention model is an IMT model recently proposed in (Li et al., 2022a), which is based on the Transformer model and has the advantage of simplicity and efficiency. We replace the image feature extraction models with video feature extraction models to fit the VMT task. The model mainly consists of the following five modules.

**Text transformer encoder**. The text transformer encoder follows the transformer encoder-decoder paradigm(Vaswani et al., 2017). The input is the source text, and the output is the text representation.

**Video feature extraction model**. The video feature extraction model's input is the video frames, and the output is the video feature.

**Selective attention**. Given the text representation $H^{\text{text}}$ and the video feature $H^{video}$, the selective attention mechanism is a single-head attention network to correlate words with video frames, where the query, key, and value are $H^{\text{text}}$, $H^{video}$ and $H^{video}$, respectively. The selective attention output $H^{\text{video}}_{attn}$ can be defined as:

$$H^{\text{video}}_{attn} = \text{Softmax}\left(\frac{QK^{\text{T}}}{\sqrt{d_k}}\right) V \qquad (1)$$

where $d_k$ is a scaling factor.

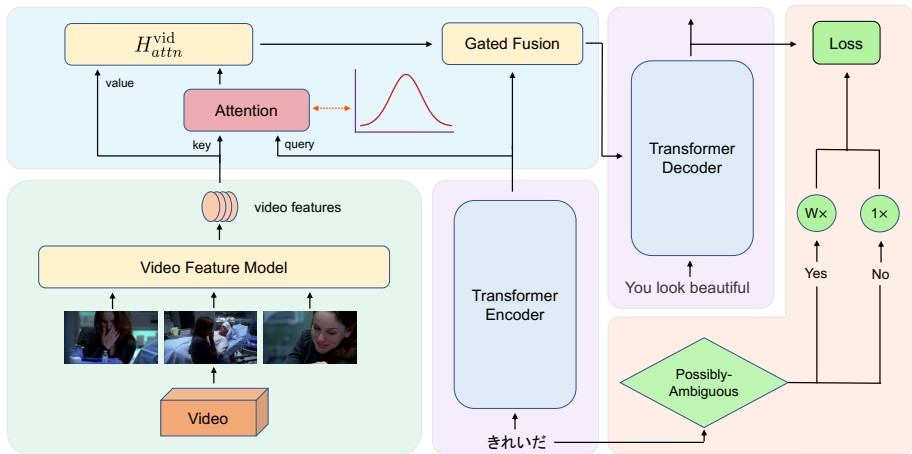

Figure 4: The SAFA model with frame attention loss (top left in red) and ambiguity augmentation (right in green). The frame attention loss uses Gaussian distribution to guide the model to pay more attention to the central frames, while the ambiguity augmentation makes the model put more weight on the data with possibly ambiguous source subtitles.

**Gated fusion** The gated fusion mechanism is a popular technique for fusing representations from different sources (Wu et al., 2021; Fang and Feng, 2022; Lin et al., 2020; Yin et al., 2020). The fused output is a weighted sum between the text representation and the selective attention output, in which the weight is controlled with the gate $\lambda$. The gate $\lambda \in [0, 1]$ and the fused output can be defined as:

$$\lambda = \text{Sigmoid}\left(UH^{\text{text}} + VH^{\text{video}}_{attn}\right) \quad (2)$$

$$H^{\text{out}} = (1 - \lambda) \cdot H^{\text{text}} + \lambda \cdot H^{\text{video}}_{attn} \quad (3)$$

where $U$ and $V$ are trainable variables. Then, the fused output $H^{\text{out}}$ is fed into the decoder.

**Transformer decoder** The transformer decoder also follows the transformer encoder-decoder paradigm. The difference is that the cross-attention block uses the fused output instead of the text representation as key and value.

The loss function $\mathcal{L}^O$ is cross entropy loss with label smoothing (Szegedy et al., 2015).

### 4.2 Frame attention loss

Unlike captions which describe the entire video clip, subtitles only occur for a few seconds in a video clip. Generally, the frames close to the subtitles are more associated with the subtitles and thus provide more information associated with the subtitles. Therefore, we hope the model can pay more attention to the frames close to the subtitles. In EVA, the subtitles occur in the center of the video clips because we have aligned the subtitles to the

videos. Inspired by (Li et al., 2020), we propose a frame attention loss that uses Gaussian distribution to guide the attention on video features.

For a random variable $X \sim \mathcal{N}\left(\mu, \sigma^2\right)$ in which $\mu = 1$ and $\sigma = 1$, we define $f_X(x)$ as the probability density function of $X$. We define $\boldsymbol{z} = [z_1, z_2, \ldots z_M]$ in which $z_1$ to $z_M$ are $M$ points equally spaced from $-a$ to $a$ and $M$ is the number of frames in a video feature. Then the frame attention loss is defined as the Kullback–Leibler (KL) divergence between the frame attention defined in Eq. (1) and the Gaussian distribution with the softmax temperature (Hinton et al., 2015) mechanism:

$$\mathcal{L}^G = \text{KL}\left(\text{Softmax}\left(\frac{QK^T}{\sqrt{d_k}}\right) \| \text{Softmax}_t\left(f_X(\boldsymbol{z})\right)\right) \quad (4)$$

where $\text{KL}()$ is the KL divergence function and $\text{Softmax}_t$ is the softmax temperature mechanism:

$$\text{Softmax}_t\left(\boldsymbol{x}\right) = \frac{\exp\left(\boldsymbol{x}/T\right)}{\sum_j \exp\left(x_j/T\right)} \quad (5)$$

where $T \in (0, \infty)$ is a temperature parameter. When $T$ gets smaller, the distribution tends to a Kronecker distribution (and is equivalent to a one-hot target vector), and the model will pay more attention to the central frames; when $T$ gets larger, the distribution tends to a uniform distribution, and the model will pay equal attention to all the frames. As we can substantially change the uniformity of the distribution by adjusting $T$, we fix $a = 3$ to reduce the number of hyperparameters.

With the frame attention loss, the loss function can be defined as:

$$\mathcal{L} = \mathcal{L}^O + \gamma \mathcal{L}^G \quad (6)$$

where $\gamma$ is a hyperparameter.

### 4.3 Ambiguity augmentation

In VMT, the video can help with translation only when the source subtitles are ambiguous. Therefore, we hope the model puts more weight on the data with ambiguous source subtitles.

For a VMT dataset $X = \{x_1, x_2, \ldots x_N\}$, $x_i$ is data consisting of a pair of parallel subtitles and a corresponding video clip. We divide the dataset into possibly-ambiguous dataset $X^a = \{x_1^a, x_2^a, \ldots x_P^a\}$ and possibly-unambiguous dataset $X^u = \{x_1^u, x_2^u, \ldots x_Q^u\}$ according to whether the source subtitle of $x_i$ is in a translation set or not, where $P$ and $Q$ are the numbers of possibly-ambiguous data and possibly-unambiguous data respectively. The loss function of the selective attention model is defined as:

$$\mathcal{L}^O = \frac{1}{N} \sum_{i=1}^{N} \mathcal{L}_{x_i} \qquad (7)$$

where $N$ is the number of data and $\mathcal{L}_{x_i}$ is the loss of the data $x_i$. Then, with ambiguity augmentation, the loss function is defined as:

$$\mathcal{L} = w \frac{1}{P} \sum_{i=1}^{P} \mathcal{L}_{x_i^u} + \frac{1}{Q} \sum_{i=1}^{Q} \mathcal{L}_{x_i^a} \qquad (8)$$

where $w > 1$ is a weight to increase the loss of possibly-ambiguous data, making the model put more weight on the possibly-ambiguous data.

## 5 Experiments

### 5.1 Settings

We conducted experiments with the Transformer configuration following (Li et al., 2022a). We train the Transformer from scratch without using external text data. For the video feature extraction model, we used CLIP4Clip (Luo et al., 2022). More details can be found in Appendix C.

We adopted BLEU (Papineni et al., 2002; Post, 2018) and METEOR (Banerjee and Lavie, 2005) as the evaluation metrics. Subtitles are essentially dialogues that are often short. Therefore we introduced the METEOR score in addition to BLEU.

For experiments in Sections 5.2 and 5.3, we repeated the experiment five times for each setting, discarded the maximum and minimum scores, and then took the average of the remaining three scores as a result. For experiments in Section 5.4, we used

the results of single experiments following the Spatial HAN model (Gu et al., 2021). Moreover, we reported the statistical significance of BLEU using bootstrap resampling (Koehn, 2004) over a merger of three test translation results.

### 5.2 Compare with SOTA

So far, there are very few VMT models. Because the Spatial HAN model (Wang et al., 2019) is not available, we use the publicly available VMT model proposed in VATEX (Wang et al., 2019) as a baseline. Existing VMT models (Wang et al., 2019; Gu et al., 2021) are mainly based on LSTM or GRU NMT model, while our model is based on the Transformer NMT model. As shown in Table 3, even the text-only model alone, which is the standard text-only Transformer model, performs much better than the previous VMT models. Therefore we focus on the comparison between our model and the text-only model. Compared to the text-only model, SAFA has comparable parameters while demonstrating significant performance gains. Specifically, for Zh-En translation, the achieves $9.99\%$ improvement in BLEU score (absolute: $1.40$) and $2.75\%$ improvement in METEOR score (absolute: $0.96$). Furthermore, for Ja-En translation, the achieves $4.94\%$ improvement in BLEU score (absolute: $1.30$) and $3.28\%$ improvement in METEOR score (absolute: $1.55$).

Furthermore, we did a context translation experiment following (Tiedemann and Scherrer, 2017) to check whether local contextual information can help disambiguate the source subtitles. We combine each subtitle with the two subtitles before and after it to construct context data and train the text-only model on context data. As shown in Table 3, the text-only model using context data performs similarly to using single subtitle data. The reason why context does not enhance translation performance may be the lack of speaker identity. The results indicate that visual information is more helpful than local contextual information in this translation task.

### 5.3 Effectiveness of the two proposed methods

In the SAFA block of Table 3, we conducted ablation studies by removing the frame attention loss (w/o Frame Attn), ambiguity augmentation (w/o Ambi Aug), and both (w/o Both). We can see that w/o Frame Attn decreases the performance more than w/o Ambi Aug, and w/o Both further decreases significantly. SAFA w/o Both (i.e., the

|  | Ja-En | | | Zh-En | | |
| Method | #param | BLEU | METEOR | #param | BLEU | METEOR |
|---|---|---|---|---|---|---|
| Text-only | 15.35M | 14.01 | 34.90 | 12.71M | 26.32 | 47.19 |
| Text-only (context) | 15.35M | 14.10 | 33.57 | 12.71M | 26.40 | 47.83 |
| VATEX | 52.41M | 12.24 | 32.07 | 36.87M | 22.15 | 44.68 |
| SAFA | 15.55M | **15.41**† | **35.86** | 12.91M | **27.62**† | **48.74** |
| - w/o Frame Attn | 15.55M | 14.98 | 35.66 | 12.91M | 27.41 | 48.59 |
| - w/o Ambi Aug | 15.55M | 15.12 | 35.55 | 12.91M | 26.81 | 48.30 |
| - w/o Both | 15.55M | 14.72 | 35.23 | 12.91M | 26.55 | 48.47 |

Table 3: Experiments on the EVA dataset. † indicates that the result is significantly better than text-only, text-only (context), and VATEX at $p < 0.01$, respectively.

| Method | #param | BLEU | METEOR |
|---|---|---|---|
| Text-only | 8.13M | 13.01 | 28.22 |
| VATEX | 52.41M | 10.50 | 25.31 |
| Spatial HAN | 57.78M | 13.19 | 28.26 |
| SAFA | 8.33M | **13.86**† | **29.09** |

Table 4: Experiments on the VISA dataset. † indicates that the result is significantly better than text-only, VATEX, and Spatial HAN at $p < 0.01$.

selective attention model) is characterized by its straightforwardness, but it does not account for the significance of the central frames where subtitles occur, nor does it allocate additional attention to ambiguous data. Therefore, it cannot take full advantage of the video information.

## 5.4 Effectiveness of SAFA on other VMT datasets

Considering the How2 dataset is an instruction subtitle dataset with long videos while VATEX is a caption dataset, we conducted additional experiments on the VISA (Li et al., 2022b) dataset that is also a subtitle dataset to test the model's generalization ability across datasets. Table 4 shows the results. While the Spatial HAN model is not available, we obtained the translation results on VISA for comparison.

It's worth noting that the Spatial HAN model is based on the winning model in the VMT Challenge competition,[1] which is a GRU model. In contrast, the VATEX model is based on LSTM. As a result, there is a significant performance gap between the Spatial HAN and VATEX models. Because VISA is relatively small and thus more appropriate for models based on GRU instead of Transformer, our text-only model performed similarly to the Spatial HAN model (Gu et al., 2021) model. We can see that the SAFA model performs significantly better than other models, although the improvements

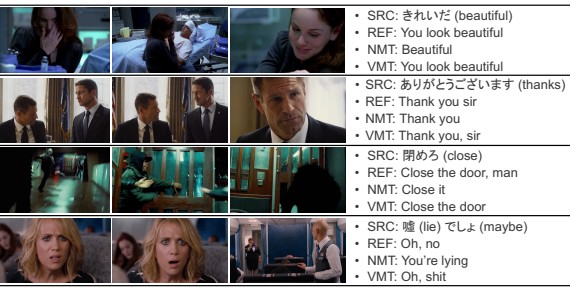

Figure 5: Qualitative examples for NMT and VMT. The VMT model is the SAFA model.

are not as large as that on EVA. On the one hand, due to the size of the VISA dataset, it only contains a small number of translation sets and a small number of possibly-ambiguous data. We think that the performance improvement of the model mainly comes from the frame attention loss method. On the other hand, the videos in VISA's evaluation set do not necessarily contribute to disambiguation.

## 5.5 Case Study

Finally, we compare several real cases to see how visual information helps translation. We choose text-only and SAFA translation results. Figure 5 shows four qualitative examples in which the VMT model uses visual information to improve the translation. Generally, as subtitles from movies and TV episodes are usually short, video can help promote the interpretation of the subtitles. In the first example of omission ambiguity, the man speaks to the woman while looking at her. Therefore, we can infer that the subject pronoun "you" is omitted in the source caption rather than an object or scene. In the second example, two men in suits are talking. Considering politeness, the address should be added. And considering the gender, the address should be "sir." In the third example, the video shows the door is closed, so the VMT model did a more generative translation. In the last example of emotion ambiguity, the woman shows a surprised

[1]https://competitions.codalab.org/competitions/24384

expression, and it is clear that the woman is not talking to someone else according to other frames. Therefore, the source subtitle should be translated as an expression of surprise rather than a statement that another person is lying. In this case, the VMT model does not translate correctly, but its translation contains emotional information. Appendix D shows the frame attention of the examples.

## 6 Conclusion

The paper introduced a new VMT dataset called EVA, which contains an extensive training set and a video-helpful evaluation set in which videos are guaranteed to be helpful for disambiguating source subtitles. In addition, we proposed a novel VMT model called SAFA that incorporates selective attention with frame attention loss and ambiguity augmentation. Experiments on EVA demonstrated that visual information and the proposed methods can boost translation performance, and SAFA performs significantly better than previous VMT models. We hope that this work will inspire further research in this field.

## 7 Limitations

The main limitations of our dataset are the following. First, the subtitles are from the OpenSubtitles dataset and provided by volunteers, which ensures the size of the dataset but results in the dataset containing some low-quality parallel subtitles. We used methods such as cross-lingual similarity to filter high-quality parallel subtitles when constructing the evaluation set but could not wholly remove low-quality subtitles. Second, each task in the crowdsourcing is only distributed to three workers due to financial constraints. If we distribute the tasks to more workers and expand the crowdsourcing scale, it is possible to obtain a higher-quality evaluation set.

## 8 Ethical Statements

We aim to facilitate MMT, so our dataset and codes will be publicly released. The subtitles utilized in this study are collected from publicly available datasets, while the videos are extracted from movies and TV episodes. We only use 10-second video clips associated with subtitles to address copyright concerns, with the audio removed. We will require all users to provide their academic affiliation as a condition to access the data. Besides, we may ask users intending to access our data to provide a self-declaration that the data is to be used solely for research purposes.

## 9 Acknowledgement

This work was supported by Google Research Scholar Award and JSPS KAKENHI Grant Number JP23H03454.

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

# A  Other Experiments

## A.1  Compare with IMT method

To compare the performance against image-based methods, we conducted IMT experiments using the Selective Attention model. In this setting, we extracted the central frame from each video clip and obtained the DETR (Dai et al., 2021) image feature of these frames to guide translations. For Ja-En translation, the BLEU score and METEOR score are 14.62 and 34.93, respectively. For Zh-En translation, the BLEU score and METEOR score are 26.36 and 47.58, respectively. Compared with the results in Table 3, the IMT method has better performance than the text-only method but not as good as the VMT method using CLIP4Clip features (w/o Both). Although the central frames are strongly associated with the subtitles, they cannot adequately capture the contextual information necessary for interpreting the subtitles. Sometimes the central frame may solely display the speaker's face, while relevant information, such as the object referred to in the subtitle, may appear before or after it.

## A.2  Effect on randomly divided evaluation set

To check the performance of the model on a test set with the original distribution, we conducted experiments on the Ja-En part of EVA with a randomly divided training set and evaluation set instead of using a video-helpful evaluation set. The size of each set was equivalent to that of EVA. The BLEU and METEOR scores of the text-only model are 13.64 and 41.58, respectively, while those of SAFA are 13.77 and 41.44, respectively. We can see that the two models have similar performance. Many samples in the randomly divided test set do not require disambiguation. Therefore, videos can not significantly help the model improve its performance.

| Method | Ja-En | | Zh-En | |
|---|---|---|---|---|
| | CIDEr | SPICE | CIDEr | SPICE |
| Text-only | 1.3167 | 6.56 | 2.0697 | 11.24 |
| Text-only (context) | 1.2571 | 6.40 | 2.0805 | **11.50** |
| VATEX | 1.0919 | 5.64 | 1.8516 | 9.39 |
| SAFA | **1.3942** | 6.80 | 2.1113 | 11.34 |
| - w/o Frame Attn | 1.3726 | 6.80 | **2.1346** | 11.49 |
| - w/o Ambi Aug | 1.3758 | **7.09** | 2.1143 | 11.12 |
| - w/o Both | 1.3380 | 6.64 | 2.1270 | 11.14 |

Table 5: Experiments on the EVA dataset.

## A.3  Experiments on other VMT evaluation set

We conducted experiments on VISA's training set and EVA's evaluation set to check the necessity of the EVA's training set. The BLEU and METEOR scores of the SAFA model are 6.77 and 25.17, respectively. Both are significantly lower than the SAFA results in Table 3. The results indicate that EVA's training set is more helpful.

## A.4  Other evaluation scores

We calculated the CIDEr (Vedantam et al., 2015) and SPICE (Anderson et al., 2016) scores for the main results. The results are shown in Table 5. Since we focus on the subtitles translation task instead of the video caption generation task, we add the results as a reference.

# B  Additional Details for Dataset

## B.1  Fix subtitle timestamps

The subtitle timestamps collected from the Open-Subtitles dataset are provided by volunteers and may not match the video. Therefore we need to align subtitles to videos to fix the timestamps. In practice, we use alass[2] to align subtitles to videos. Alass can perform subtitles alignment in two ways. One is to align subtitle files with incorrect timestamps to subtitle files with correct timestamps, such as those extracted from videos. The other is to align the incorrect subtitle file with the corresponding video using voice activity detection (VAD). We combine the two methods to do alignment and manually check the results. In this way, we make sure that the timestamps of the subtitles correspond exactly to the time when the subtitles appear.

---

[2]https://github.com/kaegi/alass

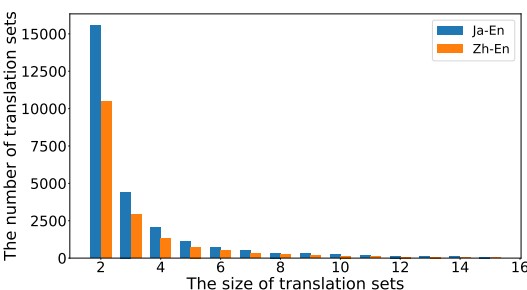

Figure 6: The size distribution of translation sets. (Until size of 15)

## B.2 Size distribution of translation sets

We examined the size distribution of translation sets in the Ja-En and Zh-En parts, separately. The result is shown in Figure 6.

## B.3 Select ambiguous translation sets

We use sent-BERT (Reimers and Gurevych, 2019) to calculate the target subtitles similarity and use the cross-lingual version of sent-BERT to calculate the parallel subtitles similarity. Sent-BERT is one of the best methods to evaluate semantic similarity.

In each translation set, for parallel subtitles with similarities higher than a threshold $T_p$, we select an ambiguous translation set if and only if the similarity between the two most different target subtitles is lower than a threshold $T_t$. Otherwise, we lower the $T_p$, and repeat the process above. Finally, we collect all the ambiguous translation sets under each $T_p$ as a result. Note that we select at most one ambiguous translation set containing two target subtitles from each translation set. Therefore the remaining data of translation sets are retained in the training set.

We set $T_t$ and $T_p$ separately. To set the threshold $T_t$, we do experiments on 100 randomly selected translation sets. We generate a ground truth by manually checking whether the source subtitle of each translation set is ambiguous and calculate the precision and recall under different $T_t$. As we do crowdsourcing later to collect final results, recall is more important than precision in this step. We set $T_t = 0.3$ with recall 0.56 and precision 0.38. Similarly, we do experiments with 200 randomly selected parallel subtitles to set $T_p$. When we only keep parallel subtitles with sentence similarity higher than 0.3, recall is 1.00 while precision is 0.90. We relax $T_p$ from 0.8 to 0.3 with 0.1 interval in sequence.

## B.4 Crowdsourcing

The crowdsourcing interface is shown in Figure 7. In the instructions, we tell workers that the subtitles occur in the center of the video clips. Because the language of most video clips is English, the workers may choose the subtitle according to the character's lip movements. Therefore we especially tell the workers to choose based on the content of the videos rather than the characters' lip movement. Moreover, we state that if the video only shows some people talking with each other and nothing special, it should not be regarded as strongly related to any subtitle.

We use qualification tests to further improve the quality of crowdsourcing. Specifically, we set qualification test tasks to check whether workers can answer them correctly. Then we only distribute large-scale tasks to workers who are good at answering this kind of task.

## C Detailed Experimental Setup

The Transformer model consists of 4 encoder and decoder layers. The input/output layer dimension is 128, and the inner feed-forward layer dimension is 256. There are 4 heads in the multi-head self-attention mechanism. We set the dropout as 0.3 and the label smoothing as 0.1.

We searched for the hyperparameters separately for the frame attention loss method, the ambiguity augmentation method, and the combination method (SAFA). For the SAFA model, we set $T = 1$, $\gamma = 0.5$, and $w = 2$. For the frame attention loss method only, we set $T = 1$ and $\gamma = 1$. For the ambiguity augmentation method only, we set $w = 2$.

Our implementation was based on Fairseq (Ott et al., 2019). For training, we used Adam Optimizer (Kingma and Ba, 2015) with $\beta_1 = 0.9$, $\beta_2 = 0.98$ and $\epsilon = 10^{-8}$. We adopted the same learning rate schedule as (Vaswani et al., 2017), where the learning rate first increased linearly for warmup $= 2,000$ steps from $1e^{-7}$ to $5e^{-3}$. After the warmup, the learning rate decayed proportionally to the inverse square root of the current step. Each training batch contained $16,000$ tokens. We also adopted the early-stop training strategy (Zhang et al., 2020) to avoid the overfitting issue.

We used Juman++ (Tolmachev et al., 2018), Stanford CoreNLP (Manning et al., 2014), and Moses (Koehn et al., 2007) to tokenize Japanese, Chinese, and English subtitles, respectively. On

**Instruction:**

Given a video (without sound) and two subtitles, is there any subtitle **strongly** related to the video **content** (eg. object, gender of characters, action, **obvious** emotion ...)? Please select the most appropriate option.
Our purpose is to find out the video clips whose content is helpful in subtitles translation, not to find out the subtitles of videos. So please choose based on the content of the videos rather than the characters' lip movement.
**If the video only shows some people talking with each other and nothing special, it should not be regarded as strongly related to any subtitle.**

**Notes:**

• Lip movements that match the subtitles are not considered as relation. (Like the first example)
• In most questions, the subtitles occur at the middle of the videos (at around 5 seconds).
• Each task contains 4 questions. Please answer all of them, then click "submit".

**Example:**

. . .

**Question 1:**

For the following two subtitles, is there any one **strongly** related to the video **content**?

A. Wait up!
B. Please!

○ None of them
○ Only subtitle A
○ Only subtitle B
○ Both of them

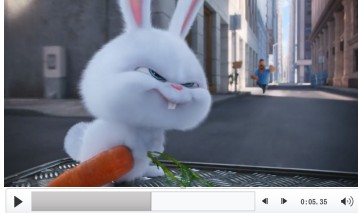

Figure 7: Crowdsourcing interface.

the EVA dataset, we mapped tokens appearing less than three times to unknown. As a result, in Ja-En translation, the Japanese vocabulary contains $38,516$ tokens while the English vocabulary contains $35,540$ tokens. In Zh-En translation, the Chinese vocabulary contains $34,860$ tokens while the English vocabulary contains $27,028$ tokens. On the VISA dataset, we used all tokens to build the vocabulary. As a result, the Japanese vocabulary contains $17,676$ tokens while the English vocabulary contains $17,732$ tokens. Therefore the number of model parameters in Tables 3 and 4 are different.

## D  Frame Attention Analysis

We show some examples of frame attention. We show four examples from the case study section (Section 5.5) and one additional example. The examples are shown in Figures 8, 9, 10, 11, and 12. In the source sentence, each token has its own attention on different frames. In the first two examples, most frames can help disambiguate the source subtitles. The model pays more attention to the first half of the video clips. In the third example, the token "閉めろ (close)" pay more attention to the fourth and fifth frames, which contain the door. Therefore, the model translates the source subtitle to " Close the door" instead of closing other things. In the fourth example, both tokens pay much attention to the third and fifth frames. Especially the fifth frame shows the surprised expression of the woman. Therefore, the source subtitle is translated as an expression of surprise. In the last example,

as the man says the sentence to a dog and he does not point to a special position, the source subtitle should be translated to "get away." In this example, the model pays more attention to the fourth, fifth, and sixth frames containing the dog.

The ambiguity of the first three examples is caused by omission, and the fourth is caused by emotion. The ambiguity of the last example is not caused by omission or emotion, and we approximately classify it as an ambiguity caused by the polysemy "いけ (go)."

- SRC: きれいだ (beautiful)
- REF: You look beautiful

- NMT: Beautiful
- VMT: You look beautiful

きれいだ (beautiful)

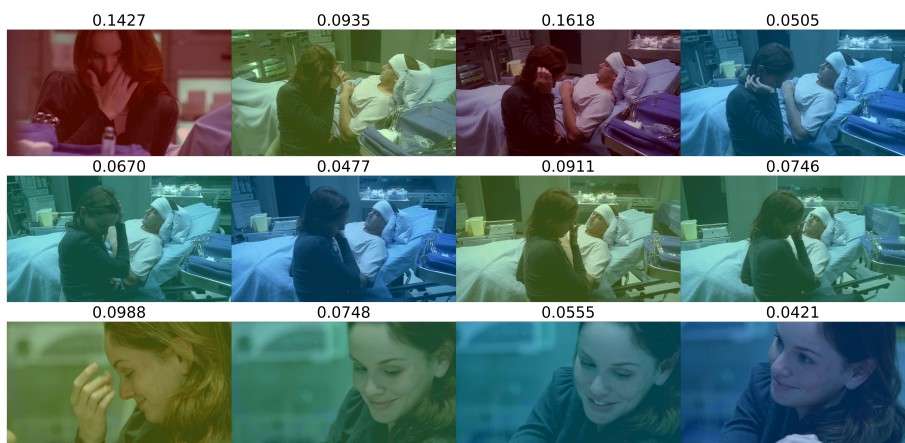

Figure 8: Frame attention of tokens in the first case study example. The attention weight of each frame is on top of the frame. The blue-to-red frame filter indicates low to high frame attention.

- SRC: ありがとうございます (thanks)
- REF: Thank you sir
- NMT: Thank you
- VMT: Thank you, sir

ありがとう (thanks)

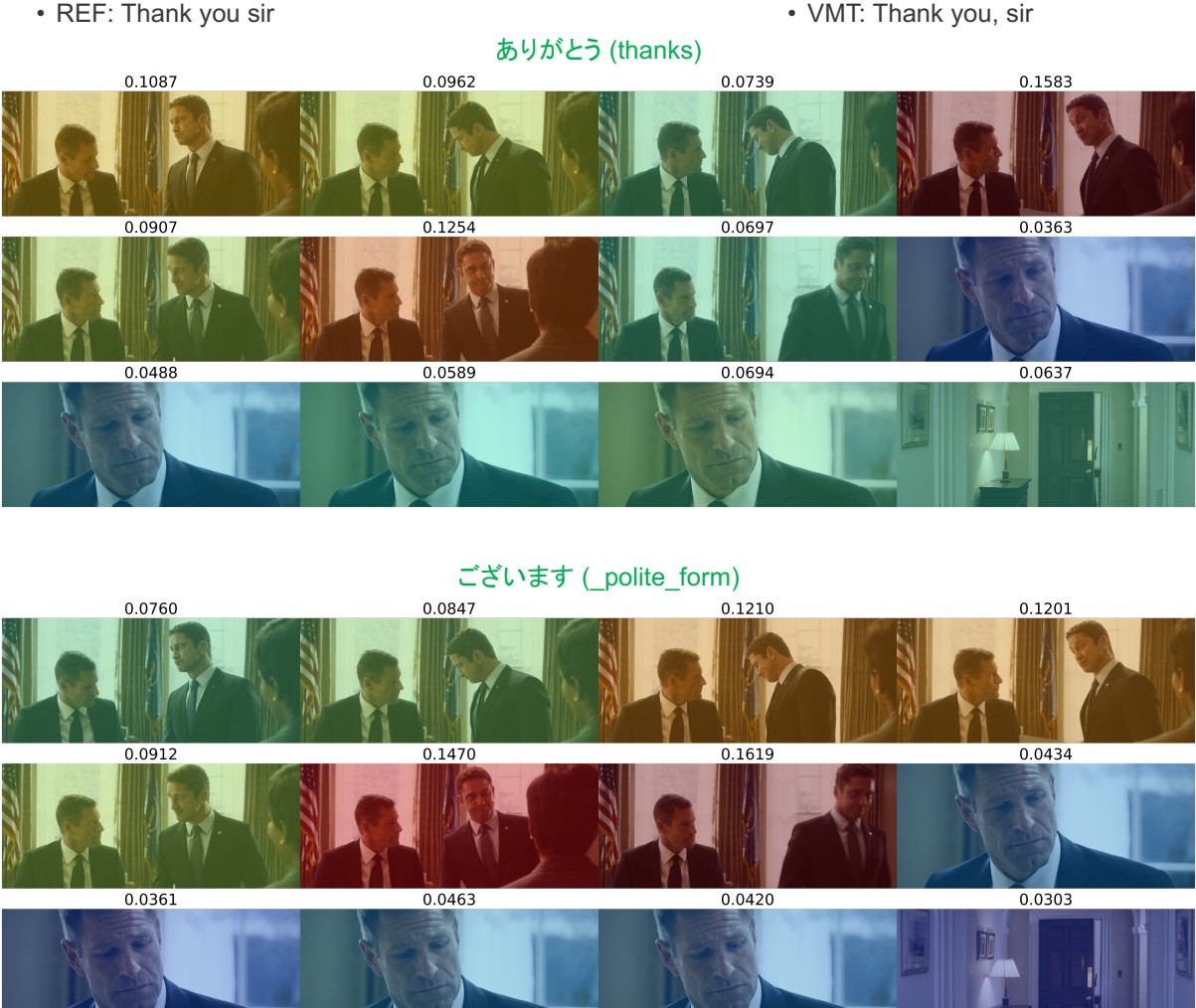

ございます (_polite_form)

Figure 9: Frame attention of words in the second case study example. The attention weight of each frame is on top of the frame. The blue-to-red frame filter indicates low to high frame attention.

- SRC: 閉めろ (close) ！
- REF: Close the door, man!

- NMT: Close it!
- VMT: Close the door!

閉めろ (close)

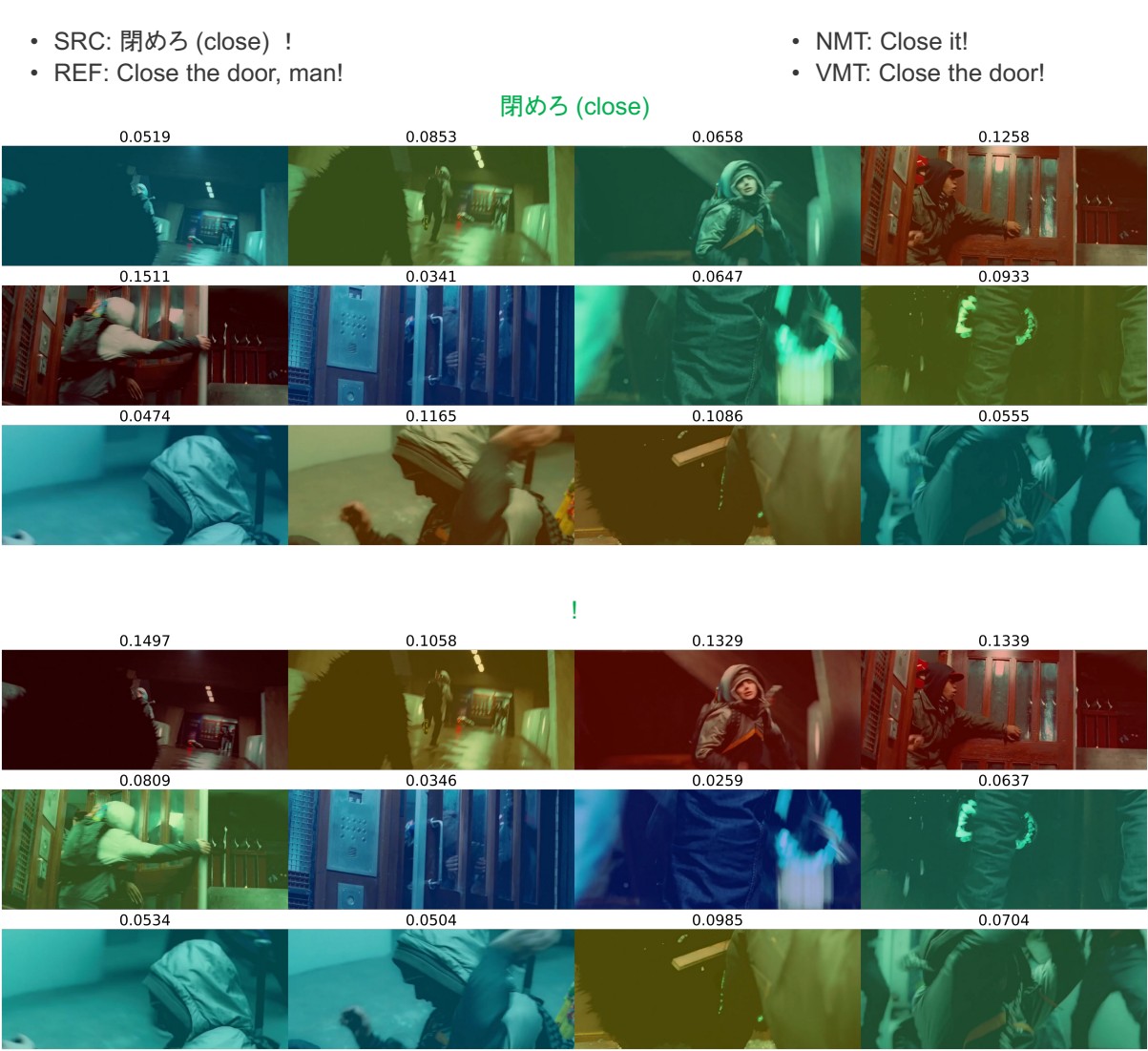

Figure 10: Frame attention of words in the third case study example. The attention weight of each frame is on top of the frame. The blue-to-red frame filter indicates low to high frame attention.

- SRC: 嘘 (lie) でしょ (maybe)
- REF: Oh, no

- NMT: You're lying
- VMT: Oh, shit

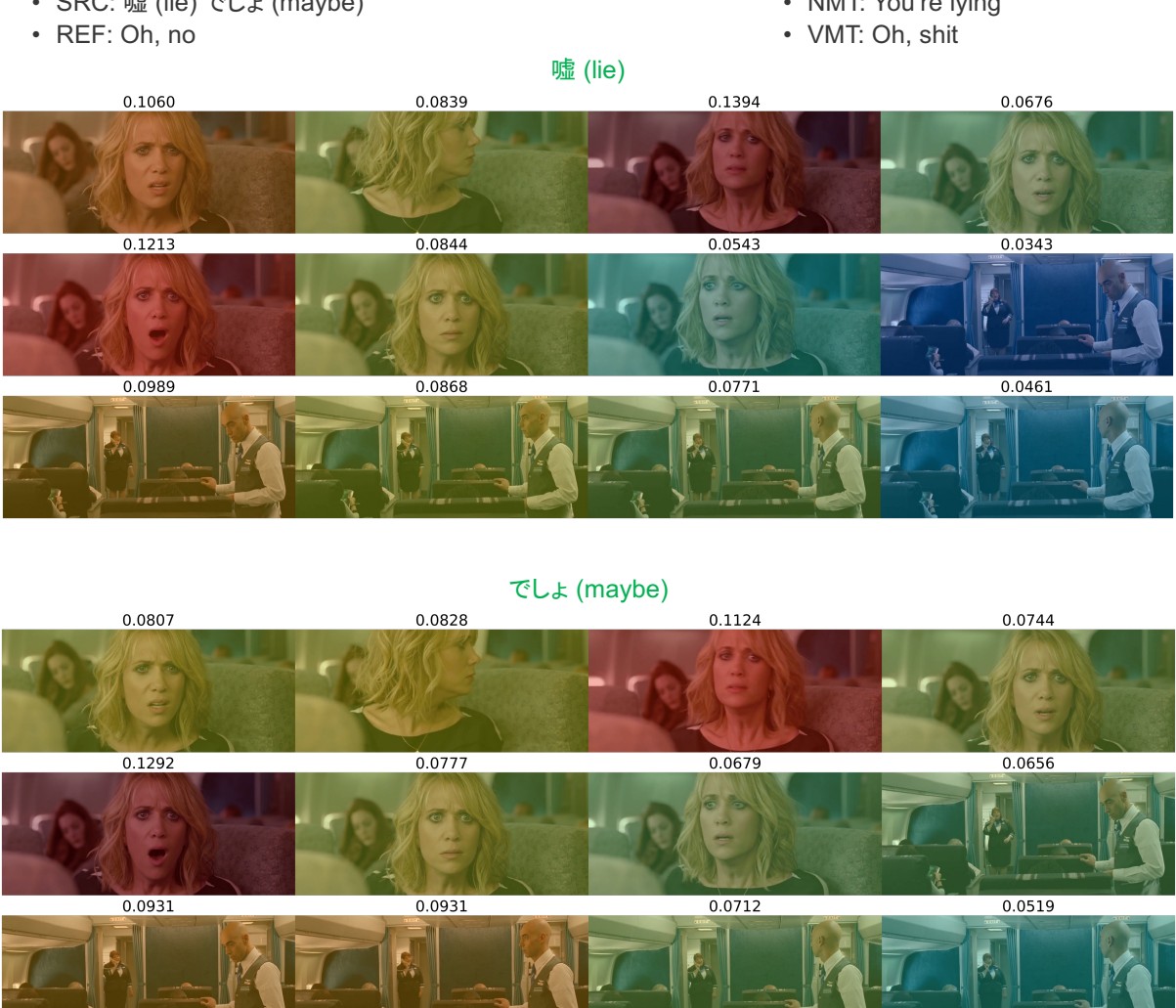

Figure 11: Frame attention of words in the fourth case study example. The attention weight of each frame is on top of the frame. The blue-to-red frame filter indicates low to high frame attention.

- SRC: あっち (there) へ (to) 行け (go)
- REF: Get away.
- NMT: Go over there.
- VMT: Go away.

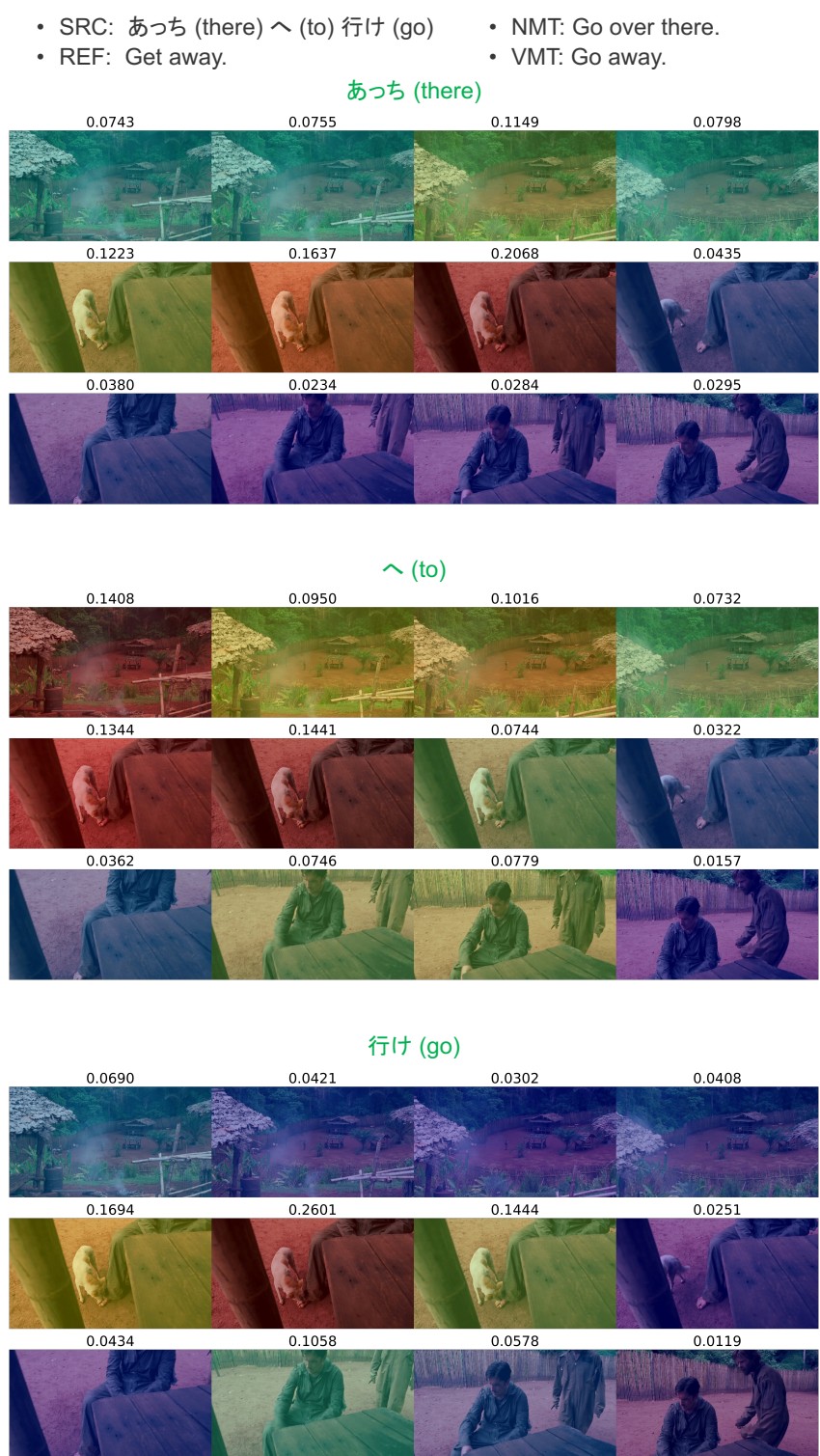

Figure 12: Frame attention of words in the fifth example. The attention weight of each frame is on top of the frame. The blue-to-red frame filter indicates low to high frame attention.