# OpenReview forum: "Video-Helpful Multimodal Machine Translation"
_EMNLP/2023/Conference — EMNLP 2023 Main_

### Official Review · Reviewer_7em6 · 2023-08-01

**Soundness:** 3

**Excitement:**

3: Ambivalent: It has merits (e.g., it reports state-of-the-art results, the idea is nice), but there are key weaknesses (e.g., it describes incremental work), and it can significantly benefit from another round of revision. However, I won't object to accepting it if my co-reviewers champion it.

**Paper Topic And Main Contributions:**

This paper argues that existing Video Machine Translation (VMT) datasets contain linguistic ambiguity, making visual information ineffective in disambiguation. To address this problem, the authors propose EVA, a large scale VMT dataset. EVA contains a video-helpful evaluation set in which subtitles are ambiguous, and videos are guaranteed helpful for disambiguation.

Furthermore, the authors propose SAFA, a VMT model with Frame attention loss and Ambiguity augmentation. Experimental results show that SAFA outperforms existing VMT models on both EVA and the previous VISA datasets.

**Reasons To Accept:**

1. The paper is well organized and easy to follow.

2. The experiments are comprehensive and solid.

    (a) I believe that the videos in the EVA dataset are somehow helpful, because according to Table 3&4, the improvement from text-only to SAFA on EVA is larger than that on the previous dataset VISA.

    (b) The ablation study in Table 3 supports the effectiveness of the proposed methods, i.e., Frame attention loss and Ambiguity augmentation.

**Reasons To Reject:**

The use of "ambiguous" and "disambiguation" is questionable, as the examples in Figure 5 seem to focus on omitted words (e.g., “Thank you, sir” vs “Thank you”, and “Close the door, man” vs “Close it”) rather than polysemy. The authors define ambiguity in Line 38, but more evidence is needed to justify this definition:
1) What proportion of EVA exhibits polysemy vs omitted words?
2) Can more examples of true polysemy be provided?

Without this justification, the framing around ambiguity and disambiguation seems inappropriate.

**Reproducibility:**

3: Could reproduce the results with some difficulty. The settings of parameters are underspecified or subjectively determined; the training/evaluation data are not widely available.

**Reviewer Confidence:**

4: Quite sure. I tried to check the important points carefully. It's unlikely, though conceivable, that I missed something that should affect my ratings.

---

> ### Author Rebuttal · Authors · 2023-08-28
>
> We sincerely thank you for your time and efforts. Please find the responses (**R**) to the concern points (**P**).
>
> **P1**: What proportion of EVA exhibits polysemy vs omitted words?
>
> **R1**: As shown in Section 3.2, the proportions of omission and polysemy are approximately 30\% and 20\%, respectively. Sometimes, instances of ambiguity arise from a combination of various factors, thereby challenging precise classification. For example, "放せ!" can be translated as both "Let me go!" and "Drop it!". This ambiguity could arise due to the polysemy "放せ!" or it could stem from the omission of the object. We will add more explanation about the classification.
>
> **P2**: Can more examples of true polysemy be provided?
>
> **R2**: Yes. For Ja-En, "取って" may be translated into "Go, fetch!" or "Take it."; "すごい人ですね" may be translated into "There are many people" or "That's a great person". For Zh-En, "放!" may be translated into "Fire!" or "Down!"; "这里" may be translated into "This way" or "Over here".

---

### Official Review · Reviewer_Dww6 · 2023-08-04

**Soundness:** 3

**Excitement:**

4: Strong: This paper deepens the understanding of some phenomenon or lowers the barriers to an existing research direction.

**Paper Topic And Main Contributions:**

This paper proposes multimodal machine translation dataset referred to as EVA (Extensive training set and Video-helpful evaluation set for Ambiguous subtitles translation) to mitigate the ambiguity in translation based on a video.

**Reasons To Accept:**

[+] Dataset contribution
[+] Writing is easy to follow
[+] Concrete problem definition
This paper is prepared for publication

**Reasons To Reject:**

[-] Provide more illustrative Figure 1 and Figure 4

**Reproducibility:**

3: Could reproduce the results with some difficulty. The settings of parameters are underspecified or subjectively determined; the training/evaluation data are not widely available.

**Reviewer Confidence:**

3: Pretty sure, but there's a chance I missed something. Although I have a good feel for this area in general, I did not carefully check the paper's details, e.g., the math, experimental design, or novelty.

---

> ### Author Rebuttal · Authors · 2023-08-28
>
> We sincerely thank you for your time and efforts. Please find the responses (**R**) to the concern points (**P**).
>
> **P1**: More illustrative Figure 1 and Figure 4
>
> **R1**: We have made improvements to Figures 1 and 4 to enhance their illustrative quality. Due to the rebuttal format, we regrettably cannot provide the revised figures here. Instead, we explain what we did to make the figures more illustrative.
>
> For Figure 1, we added the following content as an explanation: Each pair of parallel subtitles belongs to a video clip. As a translation task, the video clip can help us translate the source subtitle. The first video clip shows two women escaping, suggesting a "go" translation, while the last video clip portrays a scene involving an army, suggesting a "forward" translation.
>
> For Figure 4, we added the following content as an explanation: The frame attention loss uses Gaussian distribution to guide the model to pay more attention to the central frames, while the ambiguity augmentation makes the model put more weight on the data with possibly ambiguous source subtitles. Besides, we annotated "query,” “key,” and “value” on the figure and used a rhombus instead of the rectangle for "Possibly Ambiguous."

---

### Official Review · Reviewer_3gan · 2023-08-07

**Soundness:** 3

**Excitement:**

4: Strong: This paper deepens the understanding of some phenomenon or lowers the barriers to an existing research direction.

**Paper Topic And Main Contributions:**

This paper introduces an extensive training and test set where the video can disambiguate translations. Then a model named SAFA is designed for the disambiguation with the aid of a video.

**Questions For The Authors:**

1. Have you considered other evaluation scores, such as Cider and Spice, as they are also popular metrics in video captioning?
2. After collecting translation sets(Sec 3.1.2), what is the distribution of the size of the translation set? I think it is essential information about the ambiguity degree of the task.

**Reasons To Accept:**

1. The ambiguity issue of multimodal translation is necessary and unique. The authors propose such a large-scale dataset, which is helpful to this area.
2. The authors design two methods during training to alleviate such an issue.

**Reasons To Reject:**

The paper's definition of ambiguity (L42-L45) seems too general and vague. I think the ambiguity here mainly refers to the source texts, which must be disambiguated by the video. So the type of ambiguity, such as polysemy ambiguity (20% according to L358), is not suitable for the ambiguity in question because they, in many cases, can be determined by the context (Like "bank" in "river bank").

**Reproducibility:**

4: Could mostly reproduce the results, but there may be some variation because of sample variance or minor variations in their interpretation of the protocol or method.

**Reviewer Confidence:**

4: Quite sure. I tried to check the important points carefully. It's unlikely, though conceivable, that I missed something that should affect my ratings.

**Typos Grammar Style And Presentation Improvements:**

I don't think linguistic (L4) or language ambiguity (L72) is a suitable term to describe the kind of "ambiguity" in the paper. It tends to be confusing because linguistic ambiguity (such as structure ambiguity or polysemous ambiguity) has nothing to do with multimodal ambiguity.

---

> ### Author Rebuttal · Authors · 2023-08-28
>
> We sincerely thank you for your time and efforts. Please find the responses (**R**) to the concern points (**P**).
>
> **P1**: The definition of ambiguity
>
> **R1**: To clarify the definition, we will add "that needs multimodal information for disambiguation" at the end of the definition sentence.
>
> **P2**: Other evaluation scores
>
> **R2**: As shown in the following table, we calculated the CIDEr and SPICE scores for the main results. Since we focus on the subtitles translation task instead of the video caption generation task, we will add the results to the appendix as a reference.
>
> |                     |    Ja-En   |     Ja-En     |    Zh-En   |     Zh-En      |
> |:---------------------|:----------:|----------:|:----------:|----------:|
> | **Method**          | **CIDEr**  | **SPICE** |  **CIDEr** | **SPICE** |
> | Text-only           | 1.3167     |      6.56 |     2.0697 |     11.24 |
> | Text-only (context) | 1.2571     |      6.40 |     2.0805 | **11.50** |
> | VATEX               | 1.0919     |      5.64 |     1.8516 |      9.39 |
> | SAFA                | **1.3942** |      6.80 |     2.1113 |     11.34 |
> | - w/o Frame Attn    | 1.3726     |      6.80 | **2.1346** |     11.49 |
> | - w/o Ambi Aug      | 1.3758     |  **7.09** |     2.1143 |     11.12 |
> | - w/o Both          | 1.3380     |      6.64 |     2.1270 |     11.14 |
>
>
> **P3**: The distribution of the size of translation sets
>
> **R3**: Due to the rebuttal format, we regrettably can not show the distribution image here. Instead we show the table here. We did statistics on the distribution of the size of the translation sets. We will add the results to the appendix.
>
> | Size of Translation Sets |     2 |    3 |    4 |    5 |   6 |   7 |   8 |   9 |  10 |  11 |  12 |  13 |  14 |  15 | >15 |
> |--------------------------|------:|-----:|-----:|-----:|----:|----:|----:|----:|----:|----:|----:|----:|----:|----:|----:|
> | #Ja-En Translation Sets  | 20661 | 5742 | 2734 | 1466 | 895 | 637 | 474 | 349 | 277 | 242 | 189 | 142 | 115 | 105 | 995 |
> | #Zh-En Translation Sets  | 10459 | 2921 | 1324 |  746 | 499 | 313 | 245 | 166 | 145 | 116 |  79 |  70 |  41 |  58 | 460 |

---

### Meta-Review · Area_Chair_HPHA · 2023-09-25

**Recommendation:** 4

**Metareview:**

This paper proposes a new dataset for training and evaluation of multimodal MT. The paper focuses on cases where the visual context can aid for a better translation, eg ambiguous cases or the cases where a word is dropped. The authors have addressed almost all critical points raised by the reviewers, although a better characterization of the cases where the visual modality provides help to improve the translation would be very helpful for the paper (beyond ambiguity). Furthermore, the paper proposes a multimodal MT system to leverage the visual modality and shows that it outperforms competitive baselines. Therefore, this paper seems a a good contribution to the conference and community.

---

### Decision · Program_Chairs · 2023-10-07

**Decision:**

Accept-Main

**Comment:**

This paper proposes a new dataset for training and evaluation of multimodal MT. The paper focuses on cases where the visual context can aid for a better translation, eg ambiguous cases or the cases where a word is dropped. The authors have addressed almost all critical points raised by the reviewers, although a better characterization of the cases where the visual modality provides help to improve the translation would be very helpful for the paper (beyond ambiguity). Furthermore, the paper proposes a multimodal MT system to leverage the visual modality and shows that it outperforms competitive baselines. Therefore, this paper seems a a good contribution to the conference and community.